# *Ectothiorhodospira lacustris* sp. nov., a New Purple Sulfur Bacterium from Low-Mineralized Soda Lakes That Contains a Unique Pathway for Nitric Oxide Reduction

**DOI:** 10.3390/microorganisms11051336

**Published:** 2023-05-19

**Authors:** Irina A. Bryantseva, John A. Kyndt, Vladimir M. Gorlenko, Johannes F. Imhoff

**Affiliations:** 1Winogradsky Institute of Microbiology, Research Center of Biotechnology, Russian Academy of Sciences, 33, Bld. 2 Leninsky Ave., 119071 Moscow, Russia; 2College of Science and Technology, Bellevue University, Bellevue, NE 68005, USA; 3GEOMAR Helmholtz Centre for Ocean Research Kiel, Düsternbrooker Weg 20, D-24105 Kiel, Germany

**Keywords:** anaerobic phototrophic bacteria, purple sulfur bacteria, lamellar stacks, bacteriochlorophyll, *Ectothiorhodospira lacustris*

## Abstract

Several strains of a Gram-negative, anaerobic photoautotrophic, motile, rod-shaped bacterium, designated as B14B, A-7R, and A-7Y were isolated from biofilms of low-mineralized soda lakes in central Mongolia and Russia (southeast Siberia). They had lamellar stacks as photosynthetic structures and bacteriochlorophyll *a* as the major photosynthetic pigment. The strains were found to grow at 25–35 °C, pH 7.5–10.2 (optimum, pH 9.0), and with 0–8% (*w*/*v*) NaCl (optimum, 0%). In the presence of sulfide and bicarbonate, acetate, butyrate, yeast extract, lactate, malate, pyruvate, succinate, and fumarate promoted growth. The DNA G + C content was 62.9–63.0 mol%. While the 16S rRNA gene sequences confirmed that the new strains belonged to the genus *Ectothiorhodospira* of the *Ectothiorhodospiraceae,* comparison of the genome nucleotide sequences of strains B14B, A-7R, and A-7Y revealed that the new isolates were remote from all described *Ectothiorhodospira* species both in dDDH (19.7–38.8%) and in ANI (75.0–89.4%). The new strains are also genetically differentiated by the presence of a nitric oxide reduction pathway that is lacking from all other *Ectiothiorhodospiraceae*. We propose to assign the isolates to the new species, *Ectothiorhodospira lacustris* sp. nov., with the type strain B14B^T^ (=DSM 116064^T^ = KCTC 25542^T^ = UQM 41491^T^).

## 1. Introduction

Purple sulfur bacteria (PSB) are anoxygenic phototrophic *Gammaproteobacteria*. They are represented by the families *Chromatiaceae*, *Ectothiorhodospiraceae*, and *Halorhodospiraceae* [1]. Members of the family *Chromatiaceae* form microscope-visible sulfur globules inside the cell, whereas members of *Ectothiorhodospiraceae* and *Halorhodospiraceae* deposit sulfur globules extracellularly [2,3,4,5]. *Ectothiorhodospiraceae* include three genera: *Ectothiorhodospira*, *Thiorhodospira*, and *Ectothiorhodosinus*. The genera *Thiorhodospira* and *Ectothiorhodosinus* contain one species each (*Trs. sibirica* and *Ets. mongolicus*) [6,7,8]. The genus *Ectothiorhodospira* contains nine species with validly published names (*Ect. mobilis*, *Ect. haloalkaliphila*, *Ect. marina*, *Ect. marismortui*, *Ect. shaposhnikovii*, *Ect. vacuolata*, *Ect. variabilis*, *Ect. magna*, *Ect. salini*) and a tenth species whose name has not yet been validated (“*Ect. imhoffii*”) [1].

A large number of genome sequences of different species and strains of *Ectothiorhodospiraceae* isolated from different habitats at different times are now available [1,9]. The genome sequences of strains B14B, A-7R, and A-7Y, isolated in the late 20th century from low-salinity soda lakes in central Mongolia and southeastern Siberia (Russia), were recently analyzed to establish their taxonomic position [1,10]. Based on a comparison of genome length and G + C content (mol%) in DNA, they belong to a group of species related to *Ect. shaposhnikovii* and *Ect. vacuolata*. On the phylogenetic tree of total genomes, strains B14B, A-7R, and A-7Y form a separate branch and have ANI values <90% (75.0–89.4%) relative to any of the other strains of the genus *Ectothiorhodospira*, but high ANI values between themselves (99.0–99.9%). Therefore, they are considered to represent a new species [1].

Although these strains were isolated over two decades ago, their specific characteristics had not yet been described. Given their unique genomic placement, a more detailed characterization of these strains was warranted. This paper presents morphological and physiological characterization of strains B14B, A-7R, and A-7Y and describes the taxonomic position of the new species. Strain B14B is designated as the type of strain.

## 2. Materials and Methods

### 2.1. Bacterial Isolation

The strains studied in this work (B14B, A-7R, and A-7Y) were isolated from natural biofilm samples (thin 1–2 mm microbial mats developing on the surface of sediments) of steppe shallow-water slightly mineralized soda lakes in 1999. The color of the mats was dark green due to the development of unicellular cyanobacteria. Strain B14B was isolated from lake Bezymyannoye (salinity 1–2.0 g/L; pH 9.0) located in central Mongolia. Strains A-7R and A-7Y were isolated from lake Ostozhe (salinity 2.2 g/L; pH 9.2) located in southeastern Siberia (Russia). In these soda lakes, sodium bicarbonate makes up 80% of the total mineralization of water, while NaCl makes up the other 20%. The medium used for isolation and cultivation contained the following (per liter distilled water): NH_4_Cl 0.5 g, KH_2_PO_4_ 0.5 g, MgCl_2_∙6H_2_O 0.2 g, CaCl_2_∙2H_2_O 0.1 g, NaCl 0–5 g, KCl 0.3 g, NaHCO_3_ 3.0 g, yeast extract 0.1 g, CH_3_COONa 0.5 g, Na_2_S_2_O_3_·6H_2_O 0.5 g, Na_2_S·9H_2_O 0.5 g, vitamin solution 1 mL, trace elements solution according to Pfennig and Lippert [11] 1 mL. The vitamin solution consisted of the following components per 100 mL: thiamin 0.005 g, calcium pantothenate 0.005 g in acidic solution (0.1 N HCl, pH 3); biotin 0.002 g, 4-aminobenzoic acid 0.005 g, nicotinic acid 0.005 g, pyridoxine 0.010 g in alkaline solution (0.1 N NaOH); folic acid 0.002 g, riboflavin 0.005 g, cobalamin 0.500 g in neutral solution (deionized water). Solutions of Na_2_S·9H_2_O (10%), Na_2_S_2_O_3_·6H_2_O (10%, *w*/*v*), NaHCO_3_ (10%, *w*/*v*), yeast extract (5%, *w*/*v*), and CH_3_COONa (10%, *w*/*v*) were prepared and sterilized separately and added to the medium prior to inoculation. The final pH of the medium was 9.0. The cultures were grown at 25–35 °C under illumination (2000 lx).

Pure cultures were obtained by serial dilutions after repeated transfers of well-isolated colonies grown in 0.5–0.7% (*w*/*v*) agar medium. In liquid media, the cultures were grown anaerobically in screw-capped glass bottles at 25–35 °C and illumination of 2000 lx.

### 2.2. Morphological, Physiological, and Biochemical Characteristics

Cell morphology was studied using an Olympus BX 41 phase contrast microscope (Olympus, Tokyo, Japan). Whole cells and ultrathin sections were examined under a JEOL JEM-100C electron microscope (JEOL, Tokyo, Japan) at 80 kV. Whole cells were contrasted with 1% phosphotungstic acid (PTA). For ultrathin sectioning, the material was treated according to Ryter and Kellenberger [12], dehydrated, and embedded in Epon. The sections were placed on formvar-coated copper grids and contrasted according to Reynolds [13].

Absorption spectra of the pigments were determined within the 350–1100-nm range using an SF-56A spectrophotometer (LOMO, St. Petersburg, Russia) for both whole cells (resuspended in 50% glycerol) and in acetone extracts.

Growth at different temperature, pH, and salinity values was studied using the basic medium with variations in the tested parameters and components. The required pH was adjusted by adding HCl or Na_2_CO_3_ (18.25 or 53 g/L solutions, respectively). The concentration of organic substrates was 0.5 g/L. Biomass accumulation was determined at the stationary growth phase as OD_650_ using a KFK-3 spectrophotometer (ZOMZ, Sergiyev Posad, Russia).

Response to oxygen was determined by growth in agar stabs (0.7%, *w*/*v*) in cotton-plugged test tubes. The distance of noticeable bacterial growth from the surface of the medium was recorded after two weeks.

### 2.3. Genome Sequencing, 16S rRNA Gene Sequencing, Phylogenetic Analyses

The genome sequencing of *Ect.* strains B14B, A-7R and A-7Y was described in Imhoff et al., 2022 [1]. Average percentage nucleotide identity (ANIb) between the whole genomes was calculated using JSpecies [14]. Digital DNA-DNA hybridization (dDDH) data were obtained using the Type (Strain) Genome Server (TYGS) web server (https://tygs.dsmz.de accessed on 5 May 2023) [15]. The program used the distance formula d4 to calculate a similarity based on sequence identity.

A whole genome-based phylogenetic tree was generated using the CodonTree method within BV-BRC [16], which used PGFams as homology groups. The following genomes were used in the analysis for comparison: *Hlr. halophila* DSM 244^T^ (=SL1) (CP000544), *Mch. gracile* DSM 203^T^ (SMDC00000000), *Ect. haloalkaliphila* ATCC 51935^T^ (AJUE00000000)*, Ect. haloalkaliphila* 9902 (JAJNQR000000000)*, Ect. variabilis* DSM 21381^T^ (=WN22) (JAJNAO000000000)*, Ect. marina* DSM 241^T^ (FOAA00000000)*, Ect. mobilis* DSM 237^T^ (NRSK01000000), *Ect. marismortui* DSM 4180^T^ (FOUO00000000), *Ect. marismortui* DG9 (JAJOZD000000000), *Ect. vacuolata* DSM 2111^T^ (JAJMLZ000000000), *Ect.* sp. PHS-1 (AGBG00000000) *Ect. shaposhnikovii* DSM 243^T^ (NRSM00000000), *Ect. magna* DSM 22250^T^ (FOFO00000000), *Ect.* sp. BSL-9 (CP011994), *Ect.* sp. 9100 (JAJNAN000000000), *Ect.* sp. 9905 (JAJNAM000000000), and *Ets. mongolicus* DSM 15479^T^ (FTPK00000000). A total of 230 PGFams were found among these selected genomes using the CodonTree analysis, and the aligned proteins and coding DNA from single-copy genes were used for RAxML analysis [17,18]. iTOL version number 6.7.4 was used for tree visualization [19].

For synteny analysis, comparative genome regions were generated in BV-BRC using global PATRIC PGFam families to determine a set of genes that matched a focus gene [16]. All *Ectothiorhodospira* genomes were used in the search and compared to the B14B genome. The gene set was compared to the focus gene using BLAST and sorted by BLAST scores within BV-BRC [16]. The ‘periplasmic sensor signal transduction histidine kinase PGF_03758479’ was used as the focus gene to analyze synteny of the gene cluster.

The multiple sequence alignment for the 16S rRNA comparisons was performed using MUSCLE [20]. The evolutionary history was inferred by using the maximum likelihood method and Tamura–Nei model [21]. The bootstrap consensus tree inferred from 1000 replicates was taken to represent the evolutionary history of the taxa analyzed. Initial tree(s) for the heuristic search were obtained automatically by applying the maximum parsimony method. A discrete gamma distribution was used to model evolutionary rate differences among sites (5 categories (+*G*, parameter = 0.6304)). The rate variation model allowed for some sites to be evolutionarily invariable ([+*I*], 57.63% sites). The branch lengths were measured in the number of substitutions per site. This analysis used 16S rRNA sequences that were obtained from the genome sequences or downloaded from Genbank when genomes were not available, with the following accession numbers: AM902494 (“*Ect. imhoffii*”) and X93480 (*Ect.* sp. DSM 239). All positions with less than 95% site coverage were eliminated, i.e., fewer than 5% alignment gaps, missing data, and ambiguous bases were allowed at any position (partial deletion option). There were a total of 1464 positions in the final dataset. Evolutionary analyses were conducted in MEGA X [22], and iTOL was used for tree visualization [19]. Pairwise 16S rRNA comparisons were performed using LALIGN (https://www.ebi.ac.uk/Tools/psa/lalign/ accessed on 6 May 2023) using the genome-derived 16S rRNA sequences, and standard parameters were used.

The nucleotides of the complete DNA sequences of the genomes of the strains B14B, A-7R, and A-7Y were deposited in DDBJ/ENA/GenBank under accession no. JAJNQM000000000, JAJNQO000000000, and JAJNQN000000000, respectively [1].

The 16S rRNA gene sequences of the strains B14B, A-7R, and A-7Y were deposited with GenBank under accession numbers OQ618219, OQ618217, and OQ618218, respectively.

## 3. Results and Discussion

### 3.1. Morphology and Ultrastructure

The cell morphology of all new strains was similar. Cells were oval- to rod-shaped and sometimes slightly bent (Figure 1A,B). The cells varied from 0.5 to 1.3 µm in width and 0.7 to 2.7 µm in length. Young cells of the strains B14B and A-7R were motile by means of a tuft of polar flagella (Figure 1B). Motility was not found in strain A-7Y. Cells multiplied by binary fission. In the later stages of growth, the cells formed a mucous capsule (Figure 1C). Ultrathin sections of cells of strain B14B showed a Gram-negative type of cell wall (Figure 1D). The internal photosynthetic membranes were lamellae and located in ordered stacks, as in other species of the *Ectothiorhodospiraceae* (Figure 1C,D). Dense oval inclusions, characteristic of polyphosphates, were seen regularly in the cells (Figure 1C).

### 3.2. Pigments

Colonies on an agarized medium and cell suspensions of the studied strains B14B, A-7R, and A-7Y were bright red. The in-vivo absorption spectrum had maxima at 378, 462, 491, 594, 799, and 862 nm and shoulders (indicated in parentheses) at (510–525), (710–730), and (882–892) nm (for different strains ± 2 nm; Figure 2). The absorption spectrum of cells extracted with acetone showed peaks at 358, (390), 477, 500, 578, (700), and 772 nm (Figure 2).

The main pigment was bacteriochlorophyll *a*. Its presence on the absorption spectrum of living cells was indicated by absorption maxima at 378, 594, 799, 862 nm and a shoulder at (882–892) nm (Figure 2). On the absorption spectrum of the acetone extract, bacteriochlorophyll *a* had a major maximum at 772 nm (Figure 2). Bacteria contained carotenoids of the spirilloxanthin series, the presence of which was confirmed by the maximum at 491 and the shoulder (510–525) nm on the absorption spectrum of whole-cell pigments (Figure 2). All known representatives of the genus *Ectothiorhodospira* are characterized by the presence of bacteriochlorophyll *a* and carotenoids of the spirilloxanthin series.

### 3.3. Physiological Characteristics

Bacteria of the studied strains were anaerobic phototrophs. Photoautotrophic growth with sulfide was weak (growth yields were roughly half that of photoheterotrophically grown cultures). Photoautotrophically grown cells could be subcultured several times. Sulfide was oxidized to sulfate with the formation of S° as an oxidation intermediate, which was deposited outside the cells. Thiosulfate and sulfite were not oxidized under photoautotrophic conditions. Growth on thiosulfate and sulfite was possible only in the presence of organic substrates. Bacteria grew photoheterotrophically, with sulfide, thiosulfate and sulfite as a sulfur source for biosynthesis. Assimilatory sulfate reduction was not possible. Under microaerobic conditions in the dark, growth was only possible on organic substrates with sulfide as a source of reduced sulfur (Table 1).

Under phototrophic conditions on a medium with sulfide as a sulfur source, strain B14B used acetate, butyrate, yeast extract, lactate, malate, pyruvate, succinate, and fumarate. In the presence of arginine, aspartate, glycerol, glucose, glutamate, lactose, propionate, peptone, propanol, sucrose, sorbitol, formate, fructose, citrate, ethanol growth was not observed. Nitrogen sources were ammonium, nitrate, and urea (Table 1). Growth factors were not required, but vitamin complex and yeast extract stimulated growth.

The best growth occurred at pH 9.0; the growth range was between pH 7.5 and 10.2 (Figure 3). Growth was possible at NaCl concentrations ranging from 0% (maximum growth) to 8%; the best growth occurred at 0% NaCl (Figure 3, Table 1). The studied strains were mesophiles. Optimum growth was at 25–35 °C; the temperature limits of growth were not determined.

### 3.4. Taxonomic Position

According to the results of whole-genome sequencing, the G + C content in the DNA of the studied strains B14B, A-7R, and A-7Y was 62.9–63.0 mol% [1]. Phylogenetic analysis based on the comparison of 16S rRNA gene sequences showed that these strains belonged to the genus *Ectothiorhodospira* and revealed the greatest affinity with species *Ect. shaposhnikovii* and *Ect. vacuolata* but places the new strains on a separate clade on the 16S rRNA tree (Figure 4) [1]. Although an LALIGN analysis (https://www.ebi.ac.uk/Tools/psa/lalign/ accessed on 6 May 2023) showed the highest 16S rRNA identity with *Ect. shaposhnikovii* DSM 243^T^ (99.4%; 1530/1540 bp); *Ect. vacuolata* DSM 2111^T^ (99.0%; 1511/1526 bp), *Ect. haloalkaliphila* ATCC 51935^T^ (99.0%; 1445/1459 bp), and *Ect. magna* DSM 22250^T^ (98.8%; 1454/1471 bp), the dDDH and ANI results revealed low similarity between the studied strains and the related species that did not exceed accepted thresholds for bacterial species [14,24]. It is generally well known that whole genome-based comparisons of dDDH and ANI provide a higher resolution for species differentiation than single gene-based comparisons. dDDH values between strain B14B and the known *Ectothiorhodospiraceae* species varied from 19.7% with *Ets. mongolicus* DSM 15479^T^ to 38.8% with *Ectothiorhodospira vacuolata* DSM 2111^T^ (Table 2).

On the total genome phylogenetic tree, strains B14B, A-7R, and A-7Y formed a separate branch (Figure 5) and had ANI values <90% (75.0–89.4%) relative to any of the other strains of the genus *Ectothiorhodospira*, but high ANI values between themselves (99.0–99.9%) [1]. ANI percentages > 95% have been suggested to recognize bacteria as belonging to the same species, while bacteria with ANI <90% would be recognized in most cases as separate species [14]. The ANI with the most closely related species *Ect. shaposhnikovii* and *Ect. vacuolata* was 87.8–89.4%, which are all well below the arbitrary species cutoff of 95%.

A comparative systems analysis in BV-BRC between genomes of all known species of *Ectothiorhodospiraceae* and *Halorhodospiraceae* revealed that strains B14B, A-7R, and A-7Y had 15 unique PGFams that are not found in any other *Ectothiorhodospiraceae* genome, and they all contained a gene cluster for nitric oxide reductase (NOR), which is absent in all other species. The *norCBQD* cluster contains two NO reductase activation proteins as well as small and large NOR subunits (C and B) that show similarity to cytochrome oxidases. Nitric oxide reductase catalyzes bacterial denitrification by reducing nitric oxide (NO) to nitrous oxide (N_2_O), an obligatory step involved in the sequential reduction of nitrate to dinitrogen. The presence of *nor*CB subunits is often used as a functional marker gene to identify denitrifying bacteria [25]. In addition to the PGFam analysis, a BLAST gene search using each of the 4 *nor* genes confirmed that they were only present in strains B14B, A-7R, and A-7Y and absent in all other *Ectothiorhodospiraceae* strains. There are no known studies of nitric oxide reduction in *Ectothiorhodospiraceae*, but strains B14B, A-7R, and A-7Y at least have the genetic capacity for nitric oxide reduction. An NCBI protein BLAST search of the NorQ protein showed *Wenzhouxiangella sediminis* (81.44% identity), *Halomonas korlensis* (76.03% identity), and *Halomonas nitroreducans* (74.53% identity) as the closest named relatives. All three of these are moderately halophilic, denitrifying Gammaproteobacteria isolated from saline, alkalophilic sediments, or soils [26,27,28].

The *norCBQD* gene cluster is found upstream from a ‘periplasmic sensor signal transduction histidine kinase’ (PGF_03758479) and a ‘two-component transcriptional response regulator’ (PGF_09666309). A synteny analysis in BV-BRC showed that these two genes are only conserved in six other *Ectothiorhodospiraceae* strains (Figure 6), which, interestingly, are not the closest related to the B14B clade on the whole genome and 16S rRNA trees (Figure 4 and Figure 5). This indicates that the *norCBQD* gene cluster in the three strains, B14B, A-7R, and A-7Y may have been obtained in a more recent evolutionary event. In addition to the general genomic difference, these genetic differences are another reason to consider these strains as a separate species [1].

Based on the combination of phenotypic and phylogenetic properties of strains B14B, A-7R, and A-7Y, it was proposed to classify them as a new species for which the name *Ectothiorhodospira lacustris* sp. nov. is proposed with strain B14B^T^ (=DSM 116064^T^ =KCTC 25542^T^ = UQM 41491^T^) as type strain.

## 4. Description of *Ectothiorhodospira lacustris* sp. nov.

*Ectothiorhodospira lacustris* (*la.cus’tris*. M.L. fem. adj. *lacustris* (L. n. *lacus* a lake) inhabiting lakes, growing in lakes).

Cells are oval- to rod-shaped, sometimes slightly bent rods, 0.5–1.3 × 0.7–2.7 µm in size. Young cells are motile by means of polar flagella. Some strains are not mobile. Reproduction occurs by binary fission. Internal photosynthetic membranes are lamellar stacks. The color of cell suspensions is deep red. Photosynthetic pigments are bacteriochlorophyll *a* and carotenoids of the spirilloxanthin series. Absorption spectra of living cells have maxima at 378, 462, 491, 594, 799, and 862 nm, with shoulders at (510–525), (710–730), and (882–892) nm.

Growth occurs anaerobic in the light. Photoautotrophic growth with sulfide is weak (growth yield is roughly half that of photoheterotrophically grown cultures). Photoautotrophically grown cells can be subcultured several times. Sulfide is oxidized to sulfate, with S°, which is deposited outside the cells, as an intermediary product. Thiosulfate and sulfite are not oxidized under photoautotrophic conditions. Photoheterotrophic growth is possible with sulfide, thiosulfate, and sulfite as a sulfur source for biosynthesis under anoxic conditions. Assimilatory sulfate reduction is not possible. Chemolithotrophic growth is not possible under microoxic conditions in the dark in the presence of sulfide. Chemoheterotrophic growth is possible under microoxic conditions in the dark in the presence of sulfide.

Under phototrophic conditions and in the presence of sulfide, acetate, butyrate, yeast extract, lactate, malate, pyruvate, succinate, and fumarate are used. Arginine, aspartate, glycerol, glucose, glutamate, lactose, propionate, peptone, propanol, sucrose, sorbitol, formate, fructose, citrate, and ethanol are not used. Ammonia, nitrate, and urea are used as nitrogen sources and contain genes encoding a nitrogenase, so should be able to grow on N_2_. Growth factors are not required, but vitamin complex and yeast extract stimulate growth.

NaCl is not required for growth but up to 8% NaCl are tolerated. Optimum growth is at pH 9.0 (range pH 7.5–10.2), 0% NaCl (range 0–8%).

The bacterium is a mesophile with optimum growth at 25–35 °C. The temperature limits of growth have not been determined.

Habitat: low-mineralized soda lakes. The type strain B14B was isolated from biofilms of the shallow-water soda lake Bezymyannoye in central Mongolia (pH 9.0, mineralization 1–2 g/L, of which 80% sodium bicarbonate and 20% NaCl from the total mineralization of water).

The DNA G + C content of the species is 62.9–63.0 mol% according to genomic DNA data. The DNA G + C content of the type strain B14B is 62.9 mol% according to genomic DNA data.

The type strain B14B is deposited in the German Collection of Microorganisms and Cell Cultures GmbH (DSMZ) under accession number DSM 116064, in the Korean Collection for Type Cultures under accession number KCTC 25542, and in the Collection of Unique and Extremophilic Microorganisms under accession number UQM 41491.

The full genome DNA sequence of the type strain B14B^T^ is registered under accession no. JAJNQM000000000.

The 16S rRNA gene sequence of the type strain B14B^T^ is deposited at GenBank under accession no. OQ618219.

## Figures and Tables

**Figure 1 microorganisms-11-01336-f001:**
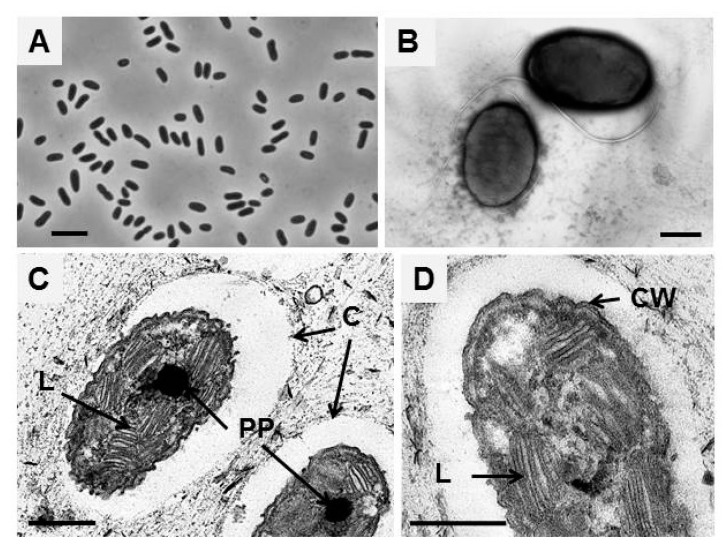
Morphology and the fine structure of strain B14B cells: (**A**) phase contrast light microscopy, (**B**) transmission electron microscopy of cells contrasted with phosphotungstic acid, and (**C**,**D**) ultrathin cell sections. Scale bar: 5 μm (**A**) and 0.5 μm (**B**–**D**). Designations: L, lamellar photosynthetic membranes; C, capsule; PP, polyphosphates; CW, cell wall.

**Figure 2 microorganisms-11-01336-f002:**
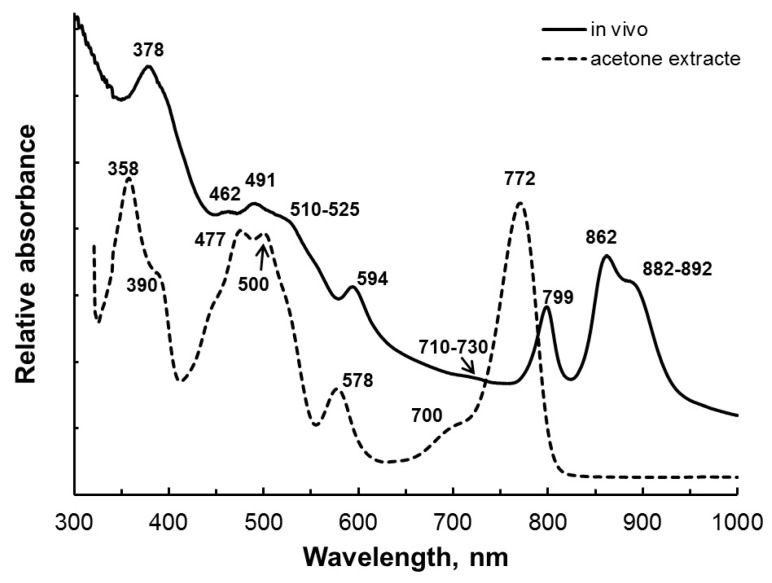
Absorption spectra of whole cells (in vivo) in glycerol (1:1, *v*/*v*) and of the acetone extract of strain B14B.

**Figure 3 microorganisms-11-01336-f003:**
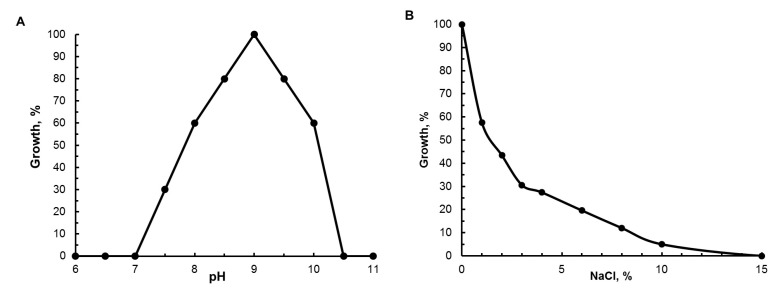
Influence of pH (**A**) and of concentrations of NaCl (**B**) on growth of strain B14B.

**Figure 4 microorganisms-11-01336-f004:**
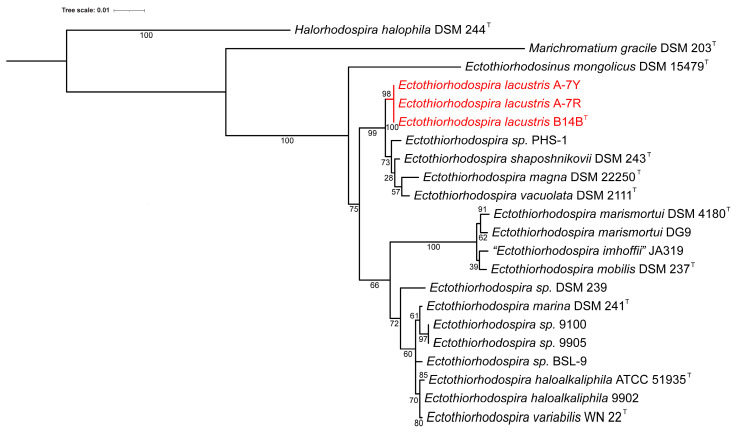
A 16S rRNA-derived phylogenetic tree for the phototrophic *Ectothiorhodospiraceae* species. *Halorhodospira halophila* DSM 244^T^
*and Marichromatium gracile* DSM 230^T^ sequences were included in the tree as outgroups. The phylogenetic tree was calculated in MegaX, and iTOL was used to draw the phylogenetic tree. Bootstrap values were generated from 1000 bootstrapping rounds. The new strains are indicated in red.

**Figure 5 microorganisms-11-01336-f005:**
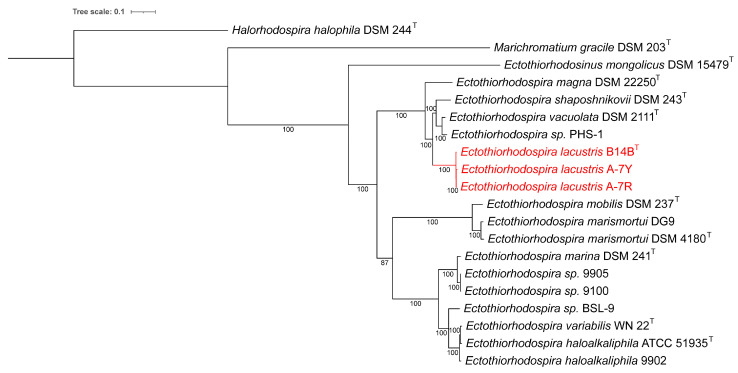
Whole genome-based phylogenetic tree of the *Ectothiorhodospiraceae*. The support values for the phylogenetic tree are generated using 100 rounds of the ‘rapid bootstrapping’ option of RaxML. The branch-length tree scale is defined as the mean number of substitutions per site, which is an average across both nucleotide and amino acid changes. The new strains are indicated in red, and *Halorhodospira halophila* DSM 244^T^ and *Marichromatium gracile* DSM 230^T^ were included in the tree as outgroups.

**Figure 6 microorganisms-11-01336-f006:**
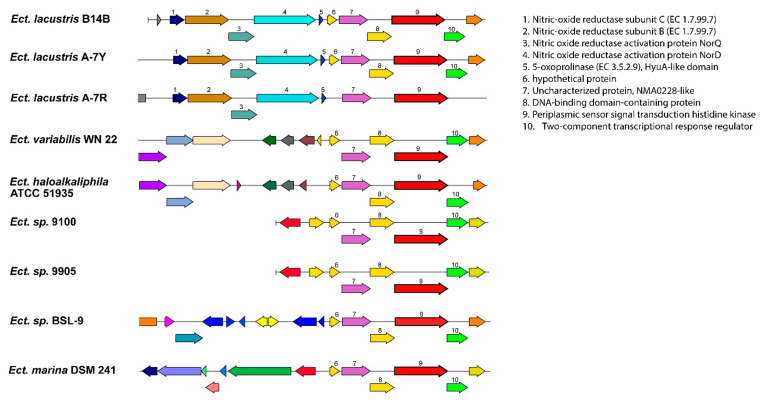
Comparison of the *norCBQD* genomic region between *Ectothiorhodospira* species. Genes are colored based on their family membership.

**Table 1 microorganisms-11-01336-t001:** Comparison of the properties of *Ectothiorhodospira lacustris* and related species.

Property	*Ectothiorhodospira* *lacustris*	*Ectothiorhodospira* *vacuolata*	*Ectothiorhodospira shaposhnikovii*	*Ectothiorhodospira magna*
Cell shape	Oval- to rod-shaped, sometimes slightly bent	Rod-shaped, sometimes slightly bent	Rod-shaped, usually slightly bent vibrios or short spirilla	Vibrioid or spirilloid
Size, µm	0.5–1.3 × 0.7–2.7	1.5 × 2–4	0.8–0.9 × 1.5–2.5	2–3.2 × 9.6–20
Motility	±, Polar tuft of flagella	+, Polar tuft of flagella	+, Polar tuft of flagella	+, Polar tuft of flagella
Gas vesicles	–	+	–	–
Internal membranes	Lamellar stacks	Lamellar stacks	Lamellar stacks	Long strands of lamellae
Color of the cell suspensions	Dark red	Pink to red	Red	Red to brownish red
Pigments	Bchl *a*, carotenoids of the spirilloxanthin series	Bchl *a*, carotenoids of the spirilloxanthin series	Bchl *a*, carotenoids of the spirilloxanthin series	Bchl *a*, carotenoids of the spirilloxanthin series
Absorption spectra of the whole cells	378, 462, 491, (510–525), 594, (710–730), 799, 862, (882–892)	378, 486, 513, 555, 593, 798, 855, (893)	378, 481, 513, 556, 593, 798, 854, (892)	373, 485–514, (545), 593, 797, 858, (885)
NaCl optimum (range), %	0 (0–8)	1–6 (0.5–10)	3 (0–7)	0.5–1.5 (0–8)
pH optimum (range)	9.0 (7.5–10.2)	7.5–9.5	8.0–8.5	9–10 (8–11)
Nitrogen sources	Ammonia, nitrate, urea, dinitrogen (genomic data)	Ammonia, dinitrogen	Ammonia, dinitrogen, nitrate, some amino acids	Ammonia
Growth factors	Not required	Not required	Not required	Yeast extract satisfies all requirements
Photoautotrophic growth	+ *	+	+	+ *
Photoheterotrophic growth	+	+	u	u
Chemolithotrophic growth	–	u	+	–
Sulfate assimilation	–	–	+	–
Substrates used:				
Hydrogen	ND	+	+	+ *
Sulfide	+	+	+	+
Thiosulfate	–	+	+	–
Sulfur	+	+	+	+
Sulfite	–	u	+	–
Organic substances	Acetate, butyrate, yeast extract, lactate, malate, pyruvate, succinate, fumarate	Acetate, malate, propionate, pyruvate, succinate, (fructose), fumarate	Acetate, butyrate, lactate, malate, propionate, pyruvate, succinate, fructose, fumarate	Acetate, malate, yeast extract *, propionate *, pyruvate *, succinate, fumarate
DNA G + C content of the species, mol%	62.9–63.0 (genomic DNA data)	61.4–63.6 (Tm)	62.0–64.0 (Tm)	59.2 (Tm)
	61.5–62.1 (Tm)			
DNA G+C content of the type strain, mol%	62.9 (genomic DNA data) 61.5 (Tm)	63.4 (genomic DNA data) 63.6 (Tm)	62.4 (genomic DNA data) 62.0 (Tm)	60.9 (genomic DNA data) 59.2 (Tm)

Symbols: +, characteristic positive or substrate utilized by most strains; –, characteristic negative or substrate not utilized by most strains; *, weak growth; (fructose), substrate utilized by some strains; ND, not determined; u, no data available; Tm, thermal denaturation. Data for reference species were taken from [1,2,23].

**Table 2 microorganisms-11-01336-t002:** Comparison of the properties of *Ectothiorhodospira lacustris* and related species. Comparison of the genomic sequences of *Ectothiorhodospira lacustris* B14B^T^ and related species based on digital DNA-DNA hybridization (dDDH) and relative G+C content differences. C.I. stands for confidence intervals.

Subject Strain	dDDH (d4, in %)	C.I. (d4, in %)	G + C% Difference
*Ectothiorhodospira vacuolata* DSM 2111^T^	38.8	[36.4–41.3]	0.52
*Ectothiorhodospira shaposhnikovii* DSM 243^T^	36.6	[34.1–39.1]	0.53
*Ectothiorhodospira magna* B7-7^T^ = DSM 22250^T^	30	[27.7–32.6]	1.97
*Thiorhodospira sibirica* A12^T^ = ATCC 700588^T^	22.3	[20.1–24.8]	6.2
*Ectothiorhodospira marina* DSM 241^T^	21.5	[19.2–23.9]	0.63
*Ectothiorhodospira variabilis* WN22^T^ = DSM 21381^T^	21.4	[19.1–23.8]	0.08
*Ectothiorhodospira marismortui* DSM 4180^T^	20.8	[18.6–23.2]	5.33
*Thioalkalivibrio sulfidiphilus* HL-EbGr7^T^	20.7	[18.5–23.1]	2.16
*Ectothiorhodospira mobilis* DSM 237^T^	20.7	[18.5–23.1]	5.42
*Thioalkalivibrio denitrificans* ALJDT = DSM 13742^T^	20.2	[18.0–22.7]	1.67
*Ectothiorhodosinus mongolicus* M9^T^ = DSM 15479^T^	19.7	[17.5–22.1]	7.29

## Data Availability

All sequence data are available in a publicly accessible repository of NCBI Genbank.

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
