# Peer review of "Ectothiorhodospira lacustris sp. nov., a New Purple Sulfur Bacterium from Low-Mineralized Soda Lakes That Contains a Unique Pathway for Nitric Oxide Reduction"

_microorganisms, 2023, doi:10.3390/microorganisms11051336_

Round 1

Reviewer 1 Report

This is a nice description of a new species of the phototrophic purple bacterial genus Ectothiorhodospira. The description is rather heavily dependent on genomics/phylogeny support since the phenotype of the new species is quite similar to that of other species of Ectothiorhodospira. I list a few things the authors could clarify in a revised version of this ms.

1. In both the abstract and on line 179, photoautotrophic growth is described as "weak".  Can a semiquantitative value be used here to better describe photoautotrophic growth? For example, were growth yields roughly half of that of photoheterotrophically grown cultures? Less than half? Could photoautotrophically grown cells be subcultured? According to the genome sequences, this species contains RuBisCO and so I'm not certain what the problem is with showing autotrophic growth. Was high sulfide levels tried or combinations of sulfide with thiosulfate? Any information along these lines would be a stronger statement than simply stating that growth was "weak".

2. Line 55. Were the biofilms distinctly pigmented? Were they actually microbial mats with other species present?

3. Line 85. It is stated that growth of the new species at various temperatures was done but no data are shown. Is the upper temperature limit for growth of this species known?

4. Lines 180-181. It is unclear what exactly is meant here concerning the growth of the new species on "thiosulfate and sulfite". Were these oxidized as photosynthetic electron donors or were they used as biosynthetic sulfur sources? The next sentence says that sulfide was the biosynthetic S source. Please clarify what is meant with thiosulfate/sulfite. If thiosulfate was oxidized as a PS electron donor, were sulfur globules produced? Also, can sulfate serve as a biosynthetic S source?

5. Lines 188-189 and Table 1. Can the new organism fix N2? All three strains contain genes encoding a nitrogenase, so they should be able to grow on N2

6. Section 3.4. The text in the first paragraph of this section seems to be at odds with the topology of the tree in Fig. 4. Specifically, it is stated that the closest relatives to the new species were Ect. vacuolata, Ect. haloalkaliphila, and Ect. magna, yet Ect. haloalkaliphila does not seem to be a close relative from its position on the tree. And how about Ect. shaposhnikovii? It seems to be the closet relative to the new species.

7. Table 2. What does the abbreviation C.I that heads column 3 stand for?

8. The nitric oxide data are interesting. Was any evidence obtained that the new species could grow anaerobically in darkness by NO reduction?

9. Description (lines 267 et seq). Add temperature range data. Also, by describing the habitat as "low mineralized", do the authors mean "low salinity"? I assume that the major salt present in the lake is NaCl? Please clarify. Also, if GPS coordinates are available to describe the exact location of the lake, these would be helpful (such are typically listed these days in isolation studies of this type).

Author Response

Dear Reviewer,

Dear Reviewer,

Thank you for considering our manuscript. We corrected the manuscript according to your recommendations and the recommendations of another Reviewer. The answers are in red.

I'm happy to improve the manuscript in response to your questions and comments.

Sincerely,

On behalf of all the authors,

Dr. Irina A. Bryantseva

Laboratory of ecology and geochemical activity of microorganisms,

Winogradsky Institute of Microbiology,

Research Center of Biotechnology of the Russian Academy of Sciences

33, bld. 2 Leninsky Ave., Moscow 119071, Russia

e-mail: bryantseva@mail.ru

Reviewr 1

Open Review

Quality of English Language

( ) I am not qualified to assess the quality of English in this paper
( ) English very difficult to understand/incomprehensible
( ) Extensive editing of English language required
( ) Moderate editing of English language
( ) Minor editing of English language required
(x) English language fine. No issues detected

Yes

Can be improved

Must be improved

Not applicable

Does the introduction provide sufficient background and include all relevant references?

(x)

( )

( )

( )

Are all the cited references relevant to the research?

(x)

( )

( )

( )

Is the research design appropriate?

(x)

( )

( )

( )

Are the methods adequately described?

(x)

( )

( )

( )

Are the results clearly presented?

(x)

( )

( )

( )

Are the conclusions supported by the results?

(x)

( )

( )

( )

Comments and Suggestions for Authors

This is a nice description of a new species of the phototrophic purple bacterial genus Ectothiorhodospira. The description is rather heavily dependent on genomics/phylogeny support since the phenotype of the new species is quite similar to that of other species of Ectothiorhodospira. I list a few things the authors could clarify in a revised version of this ms.

  1. In both the abstract and on line 179, photoautotrophic growth is described as "weak".  Can a semiquantitative value be used here to better describe photoautotrophic growth? For example, were growth yields roughly half of that of photoheterotrophically grown cultures? Less than half? Could photoautotrophically grown cells be subcultured? According to the genome sequences, this species contains RuBisCO and so I'm not certain what the problem is with showing autotrophic growth. Was high sulfide levels tried or combinations of sulfide with thiosulfate? Any information along these lines would be a stronger statement than simply stating that growth was "weak".

An explanation of the phrase "weak growth" has been added to the text. «(growth yields was roughly half of that of photoheterotrophically grown cultures).  Photoautotrophically grown cells could be subcultured several times».

In addition, other bacteria are known which contain RuBisCO but show weak growth under photoautotrophic conditions. For example, Thiorhodospira sibirica and Ect. magna.

  1. Line 55. Were the biofilms distinctly pigmented? Were they actually microbial mats with other species present?

It was a thin 1-2 mm dark green microbial mat developing on the surface of sediments. The color of the mat was determined by the development of unicellular cyanobacteria. In Lake Ostozhe in this type of biofilm (mat) were inhabited by purple sulfur bacteria morphologically similar to Thiocapsa sp..

Additions have been made to section "2.1. Bacterial isolation ".

  1. Line 85. It is stated that growth of the new species at various temperatures was done but no data are shown. Is the upper temperature limit for growth of this species known?

The studied strains are mesophiles; however, we did not determine the temperature limits of growth for them. Additions have been made to sections "3.3. Physiological characteristics" and "4. Description of Ectothiorhodospira lacustris sp. nov."

  1. Lines 180-181. It is unclear what exactly is meant here concerning the growth of the new species on "thiosulfate and sulfite". Were these oxidized as photosynthetic electron donors or were they used as biosynthetic sulfur sources? The next sentence says that sulfide was the biosynthetic S source. Please clarify what is meant with thiosulfate/sulfite. If thiosulfate was oxidized as a PS electron donor, were sulfur globules produced? Also, can sulfate serve as a biosynthetic S source?

The text has been amended to clarify the role of "thiosulfate and sulfite" in the metabolism of the new purple bacterium. It has been shown that sulfide is the electron donor during photosynthesis, in contrast to thiosulfate and sulfite, which can participate only in biosynthesis processes. Sulfate cannot serve as a source of biosynthetic sulfur. Additions have been made to sections "3.3. Physiological characteristics" and "4. Description of Ectothiorhodospira lacustris sp. nov."

  1. Lines 188-189 and Table 1. Can the new organism fix N2? All three strains contain genes encoding a nitrogenase, so they should be able to grow on N2

Growth of cultures on N2 were not tested. Additions have been made to section "4. Description of Ectothiorhodospira lacustris sp. nov. " “Contain genes encoding a nitrogenase, so should be able to grow on N2.”

  1. Section 3.4. The text in the first paragraph of this section seems to be at odds with the topology of the tree in Fig. 4. Specifically, it is stated that the closest relatives to the new species were Ect. vacuolataEct. haloalkaliphila, and Ect. magna, yet Ect. haloalkaliphila does not seem to be a close relative from its position on the tree. And how about Ect. shaposhnikovii? It seems to be the closet relative to the new species.

Ect. shaposhnikovii DSM 243T does not show up when performing NCBI BLAST comparisons. Only strain DSM2111T, which is actually Ect. vacuolata, as we described in our recent other paper, which is why the 16S rRNA % identity was originally not provided. However, we now have repeated the pairwise alignments with LALIGN, using the genome derived 16S rRNA sequences, and included Ect. shaposhnikovii in the comparison. This makes the text better aligned with the tree in Figure 4. We also further explained that ANI and dDDH comparisons have superior resolution over just single 16S rRNA comparisons for species differentiation.

  1. Table 2. What does the abbreviation C.Ithat heads column 3 stand for?

C.I. stands for confidence interval. This is now added to the table legend.

  1. The nitric oxide data are interesting. Was any evidence obtained that the new species could grow anaerobically in darkness by NO reduction?

Anaerobic growth in darkness by NO reduction was not tested. Moreover, anaerobic growth in the dark was not tested with any nitrogen compounds. Nitrogen compounds were tested as nitrogen sources under anaerobic conditions in the light.

  1. Description (lines 267 et seq). Add temperature range data. Also, by describing the habitat as "low mineralized", do the authors mean "low salinity"? I assume that the major salt present in the lake is NaCl? Please clarify. Also, if GPS coordinates are available to describe the exact location of the lake, these would be helpful (such are typically listed these days in isolation studies of this type).

In the studied lakes, sodium bicarbonate makes up 80% of the total mineralization of water, NaCl 20%.

Unfortunately, the GPS coordinates are not available. The strains B14B, A-7R, and A-7Y were isolated in 1999. The expedition members did not have the technical ability to determine GPS coordinates at that time. Additions have been made to section 4. Description of Ectothiorhodospira lacustris sp. nov..

Submission Date  20 April 2023

Date of this review  24 Apr 2023 18:32:29

Sincerely,

On behalf of all the authors,

Irina A. Bryantseva

Thank you for considering our manuscript. We corrected the manuscript according to your recommendations and the recommendations of another Reviewer. The answers are in red. I'm happy to improve the manuscript in response to your questions and comments.

Sincerely,

On behalf of all the authors,

Dr. Irina A. Bryantseva

Laboratory of ecology and geochemical activity of microorganisms,

Winogradsky Institute of Microbiology,

Research Center of Biotechnology of the Russian Academy of Sciences

33, bld. 2 Leninsky Ave., Moscow 119071, Russia

e-mail: bryantseva@mail.ru

Reviewer 2 Report

In the manuscript entitled “Ectothiorhodospira lacustris sp. nov., a new purple sulfur bacterium from low-mineralized soda lakes”, the authors presents morphological and physiologocal characterization of strains B14B, A-7R, and A-7Y and describes the taxonomic position of the new species. Overall, this manuscript is demonstrating an important and promising research direction, however part of this manuscript seems to be enhanced in much clearer discussion. Herein, it is suggested that the manuscript could be accepted for publication unless major corrections has been conducted according to the recommended points.

Required corrections:

1) Title:

The title of this manuscript needs to be modified, and to illustrate the highlights of this study.

2)  Abstract:

The abstract needs to be re-summarized to reinforce the goals of this work. Besides, the current research highlights about this study should be mentioned at the beginning. In addition, the future perspective also should be added at the end of this section.

3)  Introduction:

This section need to be rewrite and adjusted for 4-5 paragraphs. Besides, although the authors have pointed out the PSB and its family members. However, the specific advantages and its application were missing in this section, which suggested the authors should be supplemented and summarized.

4) Materials and Methods section:

Please also mention that if triplicate sets of measurements had been carried out. This is very critical for an analytical manuscript, in which the statistical analysis will be the fundamental of the conclusions. Moreover, please also indicate what is the software of data analysis in this study.

5) Results and discussion section:

It is suggested that the authors should list the advantages and comparison of this review through summarizing the current studies.

6) Description of Ectothiorhodospira lacustris sp. nov.:

Future perspective should be added in this manuscript, especially in this section.

7) Text must grammar improves and in some cases it is very weak.

8) It is suggested that English language must be polished in minor parts.

It is suggested that English language must be polished in minor parts.

Author Response

Dear Reviewer,

Thank you for considering our manuscript. We corrected the manuscript according to your recommendations and the recommendations of another Reviewer. The answers are in red.

I'm happy to improve the manuscript in response to your questions and comments.

Sincerely,

On behalf of all the authors,

Dr. Irina A. Bryantseva

Laboratory of ecology and geochemical activity of microorganisms,

Winogradsky Institute of Microbiology,

Research Center of Biotechnology of the Russian Academy of Sciences

33, bld. 2 Leninsky Ave., Moscow 119071, Russia

e-mail: bryantseva@mail.ru

Reviewr 2

Quality of English Language

( ) I am not qualified to assess the quality of English in this paper
( ) English very difficult to understand/incomprehensible
( ) Extensive editing of English language required
(x) Moderate editing of English language
( ) Minor editing of English language required
( ) English language fine. No issues detected

Comments and Suggestions for Authors

In the manuscript entitled “Ectothiorhodospira lacustris sp. nov., a new purple sulfur bacterium from low-mineralized soda lakes”, the authors presents morphological and physiologocal characterization of strains B14B, A-7R, and A-7Y and describes the taxonomic position of the new species. Overall, this manuscript is demonstrating an important and promising research direction, however part of this manuscript seems to be enhanced in much clearer discussion. Herein, it is suggested that the manuscript could be accepted for publication unless major corrections has been conducted according to the recommended points.

Required corrections:

1) Title:

The title of this manuscript needs to be modified, and to illustrate the highlights of this study.

According to the authors the title “Ectothiorhodospira lacustris sp. nov., a new purple sulfur bacterium from low-mineralized soda lakes” fully orients the reader to two main points reflected in the manuscript. 1. The article provided a rationale for describing a new species of PSB. 2. The title focused on the nature of the habitat of this new species, which is of paramount importance. We did expand the title to highlight the unique genetic differences found in these species compared to other Ectohalorhodospiraceae and Halorhodospiraceae.

2)  Abstract:

The abstract needs to be re-summarized to reinforce the goals of this work. Besides, the current research highlights about this study should be mentioned at the beginning. In addition, the future perspective also should be added at the end of this section.

 The “Abstract” does not need any major corrections. It does not exceed the established volume (179/maximum 200 words). This section briefly outlines the results of morpho-physiological studies of three strains of PSB, justifies the allocation of new isolates as a new species of the genus Ectothiorhodospira on the basis genomics/phylogeny support. However, we did also expand the abstract to highlight the specific genetic differences found in these species compared to other Ectiothiorhodospiraceae and Halorhodospiraceae. The word count now exceeds the allowed maximum (205/200).

3)  Introduction:

This section need to be rewrite and adjusted for 4-5 paragraphs. Besides, although the authors have pointed out the PSB and its family members. However, the specific advantages and its application were missing in this section, which suggested the authors should be supplemented and summarized.

The “Introduction” is 25 lines long. Meanwhile, it clearly outlines the current understanding of the taxonomy and phylogeny of the Ectothiorhodospiraceae family, taking into account recent data on the genomics of its members. We made additions and generalizations in the section 3. Results and discussion.

4) Materials and Methods section:

Please also mention that if triplicate sets of measurements had been carried out. This is very critical for an analytical manuscript, in which the statistical analysis will be the fundamental of the conclusions. Moreover, please also indicate what is the software of data analysis in this study.

 Triplicate sets of measurements were not performed when conducting physiological tests. The software which was used in the construction of phylogenetic trees was described in the Material and Methods.

5) Results and discussion section:

It is suggested that the authors should list the advantages and comparison of this review through summarizing the current studies.

 Yes, that's right. We did slightly expand upon the unique genetic differences that set these species apart from closely related species.

6) Description of Ectothiorhodospira lacustris sp. nov.:

Future perspective should be added in this manuscript, especially in this section.

 The description of the new species was made in accordance with the requirements of the minimum standards for taxonomic descriptions. The section "4. Description ..." has a certain structure of listing the properties of the described bacterium, which has been followed for many years. This section is written in accordance with established standards.

7) Text must grammar improves and in some cases it is very weak.

Minor corrections and further explanations have been provided throughout the text. This has generally improved the grammar and language, however if the reviewer has specific sentences that should be addressed, we would be glad to do so.

8) It is suggested that English language must be polished in minor parts.

Minor corrections and further explanations have been provided throughout the text. This has generally improved the grammar and language, however if the reviewer has specific sentences that should be addressed, we would be glad to do so.

Comments on the Quality of English Language

It is suggested that English language must be polished in minor parts.

Submission Date 20 April 2023

Date of this review 04 May 2023 07:57:24

Sincerely,

On behalf of all the authors,

Irina A. Bryantseva

Dear Reviewer,

Thank you for considering our manuscript. We corrected the manuscript according to your recommendations and the recommendations of another Reviewer. The answers are in red. I'm happy to improve the manuscript in response to your questions and comments.

Sincerely,

On behalf of all the authors,

Dr. Irina A. Bryantseva

Laboratory of ecology and geochemical activity of microorganisms,

Winogradsky Institute of Microbiology,

Research Center of Biotechnology of the Russian Academy of Sciences

33, bld. 2 Leninsky Ave., Moscow 119071, Russia

e-mail: bryantseva@mail.ru

Round 2

Reviewer 2 Report

The revised manuscript is clear and significantly improved. Acceptance is recommended.